META-RESEARCH ARTICLE

# Meta-analysis of variation suggests that embracing variability improves both replicability and generalizability in preclinical research

Takuji Usui[1,2¤]*, Malcolm R. Macleod[3], Sarah K. McCann[4,5], Alistair M. Senior[2☯]*, Shinichi Nakagawa[1,2☯]*

1 Evolution and Ecology Research Centre and School of Biological, Earth and Environmental Sciences, University of New South Wales, Sydney, Australia, 2 The Charles Perkins Centre and School of Life and Environmental Sciences, The University of Sydney, Sydney, Australia, 3 Centre for Clinical Brain Sciences, University of Edinburgh, Edinburgh, United Kingdom, 4 QUEST Center for Transforming Biomedical Research, Berlin Institute of Health (BIH), Berlin, Germany, 5 Charité—Universitätsmedizin Berlin Corporate member of Freie Universität Berlin, Humboldt-Universität zu Berlin, and Berlin Institute of Health, Berlin, Germany

☯ These authors contributed equally to this work.
¤ Current address: Biodiversity Research Centre, University of British Columbia, Vancouver, Canada
* usuitakuji@gmail.com (TU); alistair.senior@sydney.edu.au (AMS); s.nakagawa@unsw.edu.au (SN)

**Data Availability Statement:** All relevant data and code are available on Figshare: https://doi.org/10.6084/m9.figshare.14527317.v4

## Abstract

The replicability of research results has been a cause of increasing concern to the scientific community. The long-held belief that experimental standardization begets replicability has also been recently challenged, with the observation that the reduction of variability within studies can lead to idiosyncratic, lab-specific results that cannot be replicated. An alternative approach is to, instead, deliberately introduce heterogeneity, known as "heterogenization" of experimental design. Here, we explore a novel perspective in the heterogenization program in a meta-analysis of variability in observed phenotypic outcomes in both control and experimental animal models of ischemic stroke. First, by quantifying interindividual variability across control groups, we illustrate that the amount of heterogeneity in disease state (infarct volume) differs according to methodological approach, for example, in disease induction methods and disease models. We argue that such methods may improve replicability by creating diverse and representative distribution of baseline disease state in the reference group, against which treatment efficacy is assessed. Second, we illustrate how meta-analysis can be used to simultaneously assess efficacy and stability (i.e., mean effect and among-individual variability). We identify treatments that have efficacy and are generalizable to the population level (i.e., low interindividual variability), as well as those where there is high interindividual variability in response; for these, latter treatments translation to a clinical setting may require nuance. We argue that by embracing rather than seeking to minimize variability in phenotypic outcomes, we can motivate the shift toward heterogenization and improve both the replicability and generalizability of preclinical research.

**Funding:** AMS was supported by a Discovery Early Career Researcher Award from the Australian Research Council (ARC DECRA: DE180101520: https://www.arc.gov.au/grants/discovery-program/discovery-early-career-researcher-award-decra), and formally by a Coffey Fellowship from the University of Sydney. SN was supported by the Australian Research Council Discovery Grants (DP180100818 and DP200100367: https://www.arc.gov.au/grants/discovery-program/discovery-projects). The funders had no role in study design, data collection and analysis, decision to publish, or preparation of the manuscript.

**Competing interests:** The authors have declared that no competing interests exist.

**Abbreviations:** CAMARADES, Collaborative Approach to Meta-Analysis and Review of Animal Data from Experimental Studies; CI, credible interval; CV, coefficient of variation; HBOT, hyperbaric oxygen therapy; lnCV, log coefficient of variation; lnCVR, log coefficient of variation ratio; lnRR, log response ratio; lnSD, log standard variation; MAR, missing at random; MLMA, multilevel meta-analysis; MLMR, multilevel meta-regression; PEESE, precision-effect estimate with standard errors; PET, precision-effect test; STAIR, Stroke Academic Industry Roundtable; tPA, tissue plasminogen activator.

## Introduction

Replicability of research findings—"obtaining the same results from the conduct of an independent study whose procedures are as closely matched to the original experiment as possible," otherwise known as "Results reproducibility" [1]—is integral to scientific progress. Compelling evidence, however, suggests that non-replicability pervades basic and preclinical research [1–5]. Moreover, animal studies motivate the development of novel treatments to be tested in clinical studies, but failure to observe effects in humans which have been reported in animal studies is commonplace [6,7]. The conventional approach to preclinical experimental design has been to minimize heterogeneity in experimental conditions within studies to reduce the variability between animals in the observed outcomes [8]. Such rigorous standardization procedures have long been endorsed as the way to improve the replicability of studies by reducing within-study variability and increasing statistical power to detect treatment effects, as well as reducing the number of animals required [8,9]. This well-established notion that standardization begets replicability, however, has recently been challenged.

An inadvertent consequence of standardization is that an increase in internal validity may come at the expense of external validity [10]. By reducing within-study variability, standardization may inflate between-study variability as outcomes become idiosyncratic to the particular conditions of a study, ultimately becoming only representative of local truths [10–12]. For example, in animal studies, the interaction between an organism's genotype and its local environment (i.e., phenotypic plasticity due to gene-by-environment interactions) can result in variable and discordant outcomes across laboratories using otherwise concordant methodology [13–16]. Such inconsistent outcomes may result from distinct plastic responses of animals to seemingly irrelevant and minor, unmeasured differences in environmental conditions and experimental procedures [13–18]. Through amplifying the effects of these unmeasured variables, standardization may thus weaken, rather than strengthen, replicability in preclinical studies.

A potential counter to this "standardization fallacy" [10] then is to improve replicability by embracing, rather than minimizing, heterogeneity [10–12]. Practical solutions to enhance external validity include conducting studies across multiple laboratories to deliberately account for differences in within-lab variability [19–21], and perhaps more radically, to systematically introduce variability into experimental designs within studies [12,22,23]. Both simulation [11,14,20,21] and empirical studies [19,22,24,25] show that deliberate inclusion of more heterogeneous study samples and experimental conditions (i.e., "heterogenization") improve external validity, and hence replicability, by increasing within-study (or within-lab) variability and minimizing among-study (or among-lab) variability.

Despite the promise of heterogenization, standardization remains the conventional approach in preclinical studies [26–28]. This has been partly fuelled by Russel and Birch's [29] injunction to a "reduction in the numbers of animals used to obtain information of a given amount and precision." Consequently, within-study variability is typically treated as a biological inconvenience that is to be minimized, rather than an outcome of interest in its own right. Embracing and quantifying heterogeneity, however, may benefit preclinical science in at least 2 ways. First, through comparative analyses of the variability associated with experimental procedures, we may identify methodologies that introduce variation. As discussed above, by using methods that induce variation, one may design a deliberately heterogeneous study with greater replicability [10–12]. Second, by explicitly investigating interindividual heterogeneity in the response to drug/intervention outcomes, we may quantify the generalizability of a treatment and its translational potential. That is, a treatment with low interindividual variation in efficacy despite heterogenization is more generalizable, while a treatment with high interindividual

variation indicates the effect may be individual specific. This may be relevant in the context of personalized medicine: A treatment associated with interindividual variation in outcomes may require tailoring in its clinical use [30]. Taking these 2 points together, one could argue that an ideal trial would use a technical design that typically generated variation in disease state, which was then attenuated by a treatment of interest that might consistently (in all animals) or selectively (in some animals) improve outcome.

An illustrative case where the issues of replicability and lack of translation have been highlighted repeatedly is that of animal models of ischemic stroke [31–33]. Several systematic reviews [34,35] and meta-analyses [36–38] have questioned the propriety of experimental design and the choice of experimental procedures in stroke animal studies. The consequent recommendation for improving replicability in the field has usually been to adopt methodological procedures that minimize heterogeneity (and/or mitigate sources of bias) in phenotypic outcomes (e.g., in infarct volume or neurobehavioral outcomes) [34–38]. Furthermore, while potentially beneficial treatments have been identified in individual trials at the preclinical stage, intravenous thrombolysis remains the only regulatory-approved treatment for ischemic stroke [33,39,40]. This lack of transferable results from the preclinical to clinical stage highlights a major shortcoming for the generalizability of stroke animal models and is emblematic of translation failures generally across preclinical studies [6,7,33,34].

Using the case of rat animal models of stroke as a guiding example, we highlight how recently developed methods for the meta-analysis of variation can be used to better understand biological heterogeneity. First, through analysis of variability using the log coefficient of variation (lnCV; CV representing variance relative to the mean) in control groups, we identify methodological procedures that increase variability in outcomes. Second, we show how, through the concurrent meta-analysis of mean and variance in treatment effects using the log response ratio (lnRR; i.e., ratio of means) and log coefficient of variation ratio (lnCVR), one gains additional information about the generalizability of an intervention at the individual level. Overall, we argue that the quantification of heterogeneity in phenotypic outcomes can be exploited to improve both the replicability and translation of animal studies.

## Results

### Dataset

We obtained data for rat animal models of ischemic stroke from the Collaborative Approach to Meta-Analysis and Review of Animal Data from Experimental Studies (CAMARADES) database [41], focusing our meta-analysis on animal models that reported outcomes in infarct volume (see Materials and methods for inclusion criteria of studies). We extracted data for infarct volume from 1,318 control group cohorts from 778 studies for our analyses, investigating the effects of methodology and variability. We extracted data for the effect of treatment on infarct volume from 1,803 treatment/control group cohort pairs from 791 studies for our analyses, investigating the effects of drug treatment on interindividual variability (see Data Availability Statement section for full data and code).

### Methodology and variability

To identify methodological procedures that generated variability in disease state, we first meta-analyzed variability in infarct volume for control group animals. We quantify variability as the lnCV rather than the log of standard deviation because we found that our data showed a linear log mean–variance relationship (i.e., Taylor's law, where the variance increases with an increase in the mean [42]; S1 Fig). Overall, the coefficient of variation (CV) in infarct volume across control groups was around 23.6% of the mean (lnCV = −1.444, CI = −1.546 to −1.342,

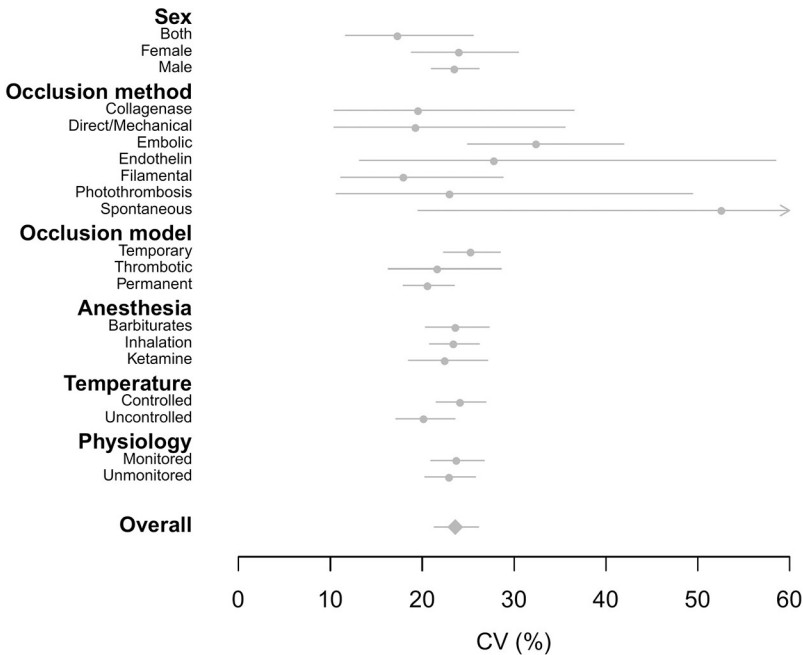

**Fig 1. The effects of methodological parameters on variability (CV) in infarct volume across control groups.** Mean estimates of unconditional (marginalized), group-specific coefficients of variation (%) are indicated as gray circles, while the overall estimate is indicated as a gray diamond. Moreover, 95% CIs are shown as gray lines and are asymmetric due to back-transformation of log coefficient of variation (lnCV) to the natural scale. Spontaneous occlusion generated the highest estimate of variability as indicated by the arrowhead. The overall and group-specific estimates were obtained from MLMA and MLMR models, respectively. The data underlying this figure can be found at https://doi.org/10.6084/m9.figshare.14527317.v4. CI, credible interval; CV, coefficient of variation; lnCV, log coefficient of variation; MLMA, multilevel meta-analysis; MLMR, multilevel meta-regression.

$\tau^2 = 0.565$; Fig 1). We found large differences in variability of infarct volume ($I^2_{total} = 93.7\%$), suggesting that sampling variance alone cannot account for differences in the reported variability across control groups (Table 1). The $I^2$ attributable to study was 49.6%, suggesting that methodological differences across studies explained some of this heterogeneity, although a moderate amount (42.9%) of $I^2$ remained unexplained (Table 1).

**Table 1. Heterogeneity ($I^2$) estimates for analyses of methodology on variability (lnCV) and drug treatment on mean (lnRR) and variance (lnCVR) in rat infarct volume.**

| Model | Total | Study | Strain | Residual (within-study) |
|---|---|---|---|---|
| *lnCV* | | | | |
| MLMA | 93.7% | 49.6% | 1.3% | 42.9% |
| MLMR | 93.3% | 46.3% | 1.7% | 45.3% |
| *lnRR* | | | | |
| MLMA | 95.7% | 54.5% | 1.7% | 39.5% |
| MLMR | 94.9% | 46.3% | 2.2% | 46.4% |
| *lnCVR* | | | | |
| MLMA | 71.2% | 38.8% | 0.9% | 31.6% |
| MLMR | 70.3% | 36.1% | 1.2% | 33.1% |

Estimates (%) are shown for MLMAs and MLMR models.

lnCV, log coefficient of variation; lnCVR, log coefficient of variation ratio; lnRR, log response ratio; MLMA, multilevel meta-analysis; MLMR, multilevel meta-regression.

We detected statistically significant differences in variability of infarct volume between various methodological approaches (Fig 1; see S1 and S2 Tables for unconditional and conditional model coefficients, respectively). Among occlusion methods, models with spontaneous occlusion produced the greatest variability in infarct volume (CV = 52.5%; lnCV = −0.644, −1.633 to 0.345), while filamental occlusion had lowest variability (CV = 17.9%; lnCV = −1.720, −2.195 to −1.244). Studies using temporary models of ischemia had higher variability in infarct volume (CV = 25.2%; lnCV = −1.377, −1.500 to −1.255) compared with permanent models. Variability was slightly but significantly lower with longer time of damage assessment (lnCV = −1.404, −1.521 to −1.288) and greater median weight of the control group cohort (lnCV = −1.366, −1.486 to −1.245).

## Drug treatment effects and interindividual variation

To quantify generalizability in drug treatment outcomes, we meta-analyzed the mean and the CV in infarct volume for the effects observed in control/experimental contrasts. We quantified the mean and interindividual variability as the lnRR and lnCVR, respectively. Overall, mean infarct volume in experimental groups was around 33.1%, smaller than in control groups (lnRR = −0.402, −0.461 to −0.343; Fig 2A), while the CV in experimental groups was around 32.4% higher than in control groups (lnCVR = 0.280, 0.210 to 0.351; Fig 2B). Overall, heterogeneity in lnRR was very high, while that for lnCVR was moderate, and moderate amounts of heterogeneity were partitioned into the study level for both (Table 1).

Both the mean and variability in infarct volume differed significantly across drug treatment groups (Fig 2; see S3 and S4 Tables for unconditional and conditional model coefficients, respectively). Treatment with hypothermia resulted in the largest reduction of mean infarct volume in experimental groups relative to controls (around 49.7% lower in experimental groups than controls; lnRR = −0.687, −0.775 to −0.599). However, hypothermia also had the most variable and inconsistent effect (i.e., intersubject variation) in reducing infarct volume, with the largest ratio of CV between experimental and control groups (interindividual variability around 60.0% higher in experimental groups compared with controls; lnCVR = 0.470, 0.349 to 0.591). In contrast, environmental treatments were the least effective in reducing mean infarct volume (around 7.3% greater in experimental groups than controls; lnRR = 0.071, −0.166 to 0.308). Hyperbaric oxygen therapy (HBOT) has the least variable and most consistent effect on infarct volume (variability around 45.3% less in experimental groups relative to controls; lnCVR = −0.603, −1.483 to 0.277).

Thrombolytics, which include the only regulatory-approved treatment (i.e., tissue plasminogen activator; tPA [33]), reduced mean infarct volume by around 29.6% in experimental relative to control groups (lnRR = −0.351, −0.446 to −0.256). The CV across experimental groups for thrombolytics was around 17.4% higher than control groups (lnCVR = 0.160, 0.031 to 0.289), but it is notable that this increased intersubject variability is much less than that seen with hypothermia. Through quantifying variability in drug treatment outcomes, we propose that treatments be considered generalizable if they reduced mean infarct volume and concurrently show low interindividual variability (i.e., negative lnRR and lnCVR estimates; Fig 3). Drug treatments that on average reduced infarct volume but had variable and inconsistent effects (i.e., had negative lnRR and positive lnCVR estimates; Fig 3) are ungeneralizable but might be appropriate for clinical exploitation in selected patients [30,43]. Conversely, the least successful treatments can be identified as those that consistently do not reduce mean infarct volume (i.e., positive lnRR and lnCVR estimates; Fig 3). We explored whether the sex of groups used in experiments affected lnRR or lnCVR (see Materials and methods for multilevel meta-regression [MLMR] model parameters), but differences in mean or variability of infarct

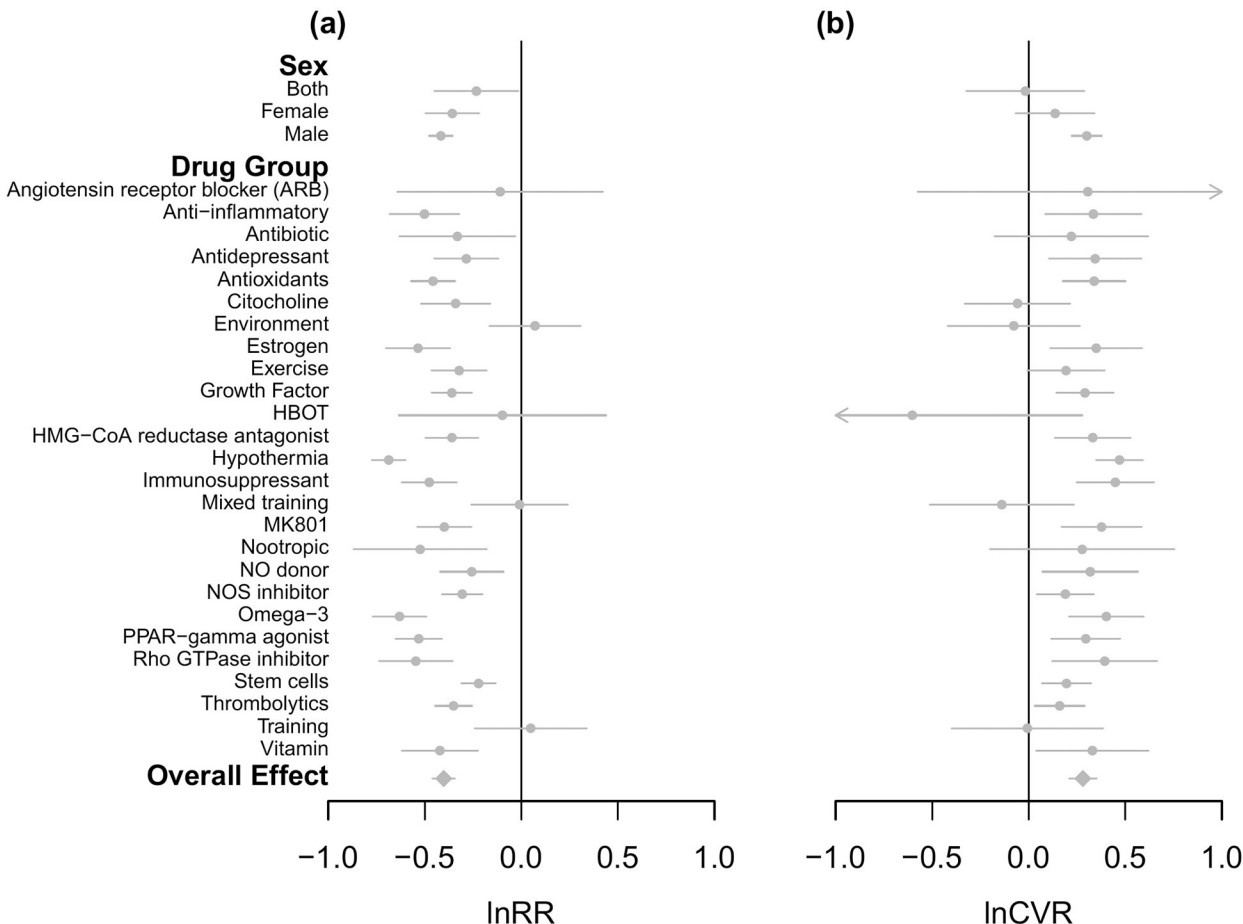

**Fig 2. The effects of drug treatments on the difference in (a) mean (lnRR); and (b) variability (lnCVR) in infarct volume across control and experimental rat groups.** Mean estimates of unconditional (marginalized), group-specific effects are shown as gray circles, while the overall estimate is indicated by the gray diamonds. Moreover, 95% CIs are shown as gray lines. Negative lnRR estimates indicate that mean infarct volume is smaller in experimental versus control rats. Negative lnCVR estimates show that interindividual variability in infarct volume is smaller in experimental versus control rats (e.g., HBOT indicated by left-pointing arrowhead), while positive lnCVR estimates show that variability in infarct volume is greater in experimental versus control rats (e.g., ARBs indicated by right-pointing arrowhead). The overall and group-specific estimates were obtained from MLMA and MLMR models, respectively. The data underlying this figure can be found at https://doi.org/10.6084/m9.figshare.14527317.v4. ARB, angiotensin receptor blocker; CI, credible interval; HBOT, hyperbaric oxygen therapy; HMG-CoA, β-Hydroxy β-methylglutaryl-CoA; lnCVR, log coefficient of variation ratio; lnRR, log response ratio; MLMA, multilevel meta-analysis; MLMR, multilevel meta-regression; NO, nitric oxide; NOS, nitric oxide synthase; PPAR, peroxisome proliferator–activated receptor.

volume did not vary significantly between female and male cohorts (see S5 and S6 Tables for contrast model estimates for sex effects).

## Discussion

We propose that the current failures in replicability and translation of preclinical trials may be due, at least in part, to the way studies are designed and assessed, which is to minimize within-study variation and ignore heterogeneity in outcomes [8,9,26–28]. Here, we have illustrated the potential utility of embracing such heterogeneity, through meta-analyzing variability (relative variance or CV) in outcomes for rat animal models of stroke. First, by estimating the variability generated by different methodological designs applied to control animal groups, we have identified procedures that generate variability in disease states (Fig 1). Second, we have, for the first time, quantified both the efficacy and stability (i.e., changes in the mean and

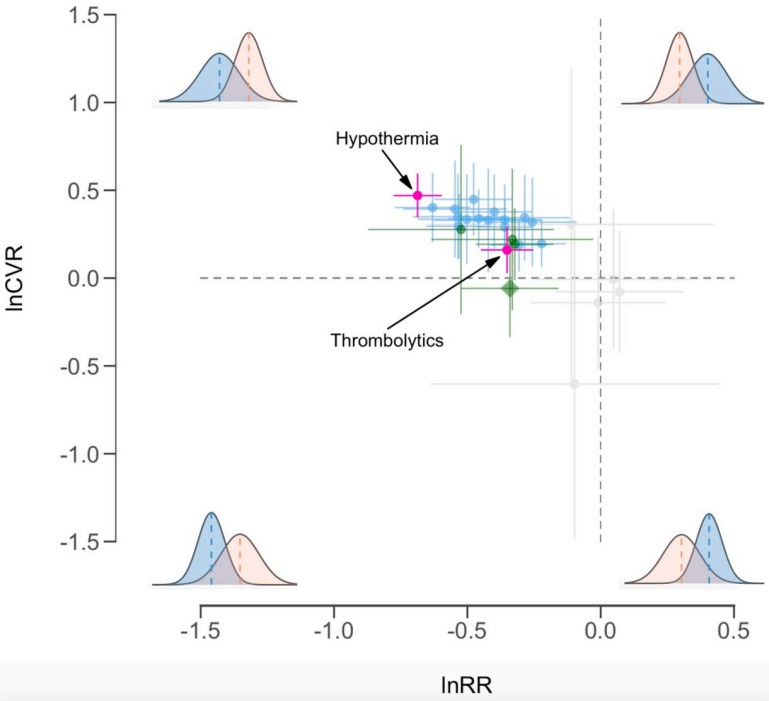

**Fig 3. Categorization of treatment effects based on mean efficacy (lnRR) and interindividual variability in efficacy (lnCVR).** Estimates (circles) represent unconditional (marginalized), treatment-specific means (lnRR), variability (lnCVR), and their 95% CIs (solid lines) obtained from MLMR models. Treatments that significantly reduce infarct volume (negative lnRR) without significantly affecting the variation are highlighted green, with citicoline indicated by a diamond as the only treatment to significantly reduce infarct volume and also have a negative point estimate of lnCVR. Treatments that significantly reduce infarct volume and increase interindividual variability (positive lnCVR) are highlighted blue. The effects of hypothermia (most negative and positive mean and variability estimates, respectively) and thrombolytics (which include the only regulatory-approved treatment) are highlighted in pink. Histograms show the relationship of the mean and variance in infarct volume between control (orange) and treatment (blue) groups in each quadrant of the graph. The data underlying this figure can be found at https://doi.org/10.6084/m9.figshare.14527317.v4. CI, credible interval; lnCVR, log coefficient of variation ratio; lnRR, log response ratio; MLMR, multilevel meta-regression.

variance, respectively) of stroke treatments applied to the experimental animal models (Figs 2 and 3), identifying potential treatments that may be generalizable versus those that require tailoring. We further discuss these results below in the context of their implications for improving the replicability (also defined as "Results reproducibility" [1]) and generalizability of preclinical studies.

## Generate variability through methodology

Among stroke animal models, studies may differ in the design of a number of parameters, including the genetic composition of animals (e.g., the sex and strain of rats used [32,44]) as well as laboratory and operational environments (e.g., methods for stroke induction, the duration of ischemia, and the type of anesthesia used [37,38,45]). However, an impediment to heterogenization is that we have not previously had reliable estimates for which methodological parameters may be most successful in generating variability in phenotypic outcomes [15]. Our results therefore quantify heterogeneity and rank the experimental factors that can generate variability in disease state into animal models so that we can most efficiently capture heterogeneity in experimental design.

Our analyses of operational factors reveal that heterogeneity in outcomes may be induced by incorporating spontaneous (CV = 52.5%), embolic (CV = 32.3%), and endothelin (CV = 27.8%) methods of occlusion. Temporary models of occlusion also generate significantly more variability in disease state than permanent models (CV = 25.2% and 20.5%, respectively). Where choices permit, we suggest that these operational design considerations are a valuable approach for introducing variability into animal models, in conjunction with more familiar proposals to diversify the laboratory environment (e.g., through differences in animal housing conditions and feeding regimens [16,19]). Depending on the type and purpose of study, such operational and laboratory design considerations that increase heterogeneity in outcomes through environmental effects may be especially valuable when variability cannot be introduced through the animal's genetic composition (e.g., for studies that are interested in sex-specific [46,47] or strain-specific outcomes [44,48]).

Considering genetic factors, proposals to include more heterogeneous study samples recommend the inclusion of both sexes over just male or female animals [49–51], as well as the use of multiple strains of inbred mice and rats (or even, multiple species) [52,53]. Recent meta-analyses of variability in male and female rodents show that males may be as or more variable than females in their phenotypic response [54,55]. We also find that male (CV = 23.5%) and female (CV = 23.9%) rats generate quantitatively equal amounts of variability. Counterintuitively, however, we find that studies that used both sexes produce the most consistent outcomes (CV = 17.3%; see S1 Table for full, unconditional model coefficients). We caution that a moderate amount of the total heterogeneity remained unexplained (i.e., residual variation; Table 1). Thus, these outcomes of sex on estimates of variability may be due to confounding effects of unaccounted for differences in experimental design. We therefore emphasize the importance of considering both genetic and environmental parameters for effective heterogenization of studies [56,57].

An alternative approach to heterogenization of experimental designs within studies is to introduce variability by conducting experiments across multiple research laboratories (i.e., multi-laboratory approach) [20,24,58]. Importantly, such an approach inherently captures "unaccounted" sources of variability in experimental conditions that are difficult to systematically manipulate within a single-center study [16,19]. We argue that, especially where logistical constraints may hinder multi-laboratory approaches (e.g., for earlier, basic, and exploratory studies), introducing heterogeneity within studies may provide the most practical alternative [23]. Indeed, by meta-analyzing the variability introduced by differences in experimental methodology across studies, we can begin to find ways in which to heterogenize single studies in order to best capture the variation that exist across laboratories and studies [16,20].

Systematically introducing variability into a system comes at the cost of reduced statistical sensitivity [8,9] and necessitates larger studies [8,26,29]. While in the long-term increased replicability may reduce waste and outweigh the initial costs, these economic and ethical costs must, of course, be minimized. This can be done by identifying from a spectrum of all available methodological choices the most efficient means of introducing heterogeneity within experiments (Fig 1, S1 Table). For some methodological aspects such as operational factors, this will mean replacing current methodologies with choices that induce greater variability in baseline and control group outcomes (e.g., by changing methods of occlusion). For other design parameters that may traditionally be standardized such as genetic or lab environmental factors, this will mean deliberately incorporating these types of heterogeneity in a systematic manner, by including levels of these categorical (e.g., strain) or continuous (e.g., time of assessment) variables using a randomized block or fully factorial design (i.e., "controlled heterogenization" [59]). Regardless of the manner in which heterogeneity is incorporated, however, it is necessary to quantify the amount of variability that different experimental designs introduce, with

the aim that researchers can then make informed decisions about how to most efficiently incorporate heterogeneity into study design [14–16,20]. Identifying sources of variability through meta-analysis of variance in existing animal data as we have done here is the most practical and economic way of establishing this much needed knowledge base.

Our analysis is not the first to assess the effects of experimental methodology on variation in disease state in rodent models of stroke [37 38]. Ström and colleagues [37] investigated similar components of experimental design on variation in infarct volume in rats. There are a number of methodological differences between their analyses and ours (e.g., differences in size of dataset and use of formal meta-analytic models). Despite these differences, our quantitative results are largely concordant. Where we differ substantially is in the interpretation of what is a desirable outcome. For example, Ström and colleagues [37] concluded that intraluminal filament procedures provide optimal occlusion methods as they generate minimal variation in disease outcome and maximize statistical power. Our analyses also identify that filament methods have low variation (CV = 17.9%); however, we argue that these gains in statistical power come at the cost of reduced replicability.

We attempted to provide formal statistical support for the hypothesis that heterogeneous methods result in more repeatable treatment effects. We used a second-order meta-regression to assess whether the amount of variation (lnCV) in disease states induced by occlusion methodology predicts heterogeneity in effect sizes for drug treatments using those methods (quantified as ln$H$ [60]). As predicted, there is a negative relationship (slope = −0.876, −2.047 to 0.295; $P$ = 0.142; S2 Fig), suggesting that methodologies that induce greater variability in baseline disease states are associated with more consistent treatment outcomes. We note, however, that our slope estimate is statistically nonsignificant and that our analysis was based on a small number of methodological groups ($N$ = 7) with an unbalanced distribution of drugs/rat strains across those groups (see S7 Table for analysis details and full model results). Nonetheless, our results are encouraging, and we are excited to see further studies formally quantify the relationship between variability induced by methodological procedures and replicability in reported outcomes. Meta-analyses that quantify both variability in control and treatment outcomes, as done here, provide a useful approach for quantifying the relationship between methodological heterogenization and replicable outcomes.

## Quantify variability to improve drug translation

Our second approach of simultaneously assessing both the mean and variation in treatment outcomes allows us to place potentially useful treatments into 2 distinct categories for further exploration: (1) beneficial and generalizable interventions, which are those that consistently reduce infarct volume across individuals; and (2) beneficial but non-generalizable interventions, which on average reduce infarct volume but result in large interindividual heterogeneity in outcomes. This latter group could even include treatments that do not necessarily reduce mean state, but have a large enough variance response to be beneficial to some [30,43,61].

Overall, we find that the stroke treatments in our dataset are usually effective, reducing infarct volume on average by 33.1% compared with controls. Out of these effective treatments, we identify 4 treatments that significantly reduced infarct volume but did not induce significant differences in the CV across experimental and control groups (green highlights in Fig 3). Nootropic treatments reduced infarct volume on average by 40.8%, while citicoline, antibiotic, and exercise treatments reduced infarct volume by around 27.5% to 28.8% compared with control groups. None of these treatments were estimated to significantly affect the CV, although estimated effects ranged from 5.7% smaller in experimental relative to controls for citicoline (highlighted with a triangle symbol in Fig 3) to 21.3% to 31.9% greater for the other

treatments. We emphasize that these treatments may potentially be more generalizable in that the outcomes of these treatments are on average favorable and are relatively consistent at the individual level [33,34].

Second, we identify a handful of effective treatments that on average reduce infarct volume, but also generate significant amounts of variability in experimental groups (blue highlights in Fig 3; see S3 Table for rank order of unconditional estimates in mean and CV across treatments). Of particular interest to note is that while thrombolytics significantly increase variability in experimental groups relative to controls, they are still relatively consistent in reducing mean infarct volume (on average reducing infarct volume by 29.6%, while the CV in experimental groups is only 17.4% greater than controls). Out of treatments that significantly reduce mean infarct volume, thrombolytics rank second in terms of its consistency in effect, with overlapping confidence intervals in their effects on the CV with those of citicoline (Fig 3).

On the other hand, hypothermia is much more effective in reducing infarct volume (on average reducing infarct volume by 49.7%) but is the least consistent in doing so, estimating the greatest CV (60.0% greater in hypothermia treated groups than concurrent controls). Interestingly, efforts to exploit hypothermia for stroke in clinical trials have so far failed to identify a patient group who might reliably benefit [62]. Other treatments that greatly reduce average infarct volume while increasing the variation include, for example, omega-3, rho GTPase inhibitors, and estrogen treatments. As such, while these treatments confer a mean beneficial effect, this effect may not be generalizable across animals. Any future translation into clinical trials would require tailoring with effort put in to predicting response at the individual level [30]. To our knowledge, such tailoring has not been attempted because a treatment with high variability (inconsistency) is less likely to be statistically significant and pass the preclinical stage (even if it does improve a disease state) [30,43,61,63]. Our study represents the first meta-analyses to quantify both the efficacy and consistency of treatment effects in animal models. We believe that this approach will forge new opportunities for improving the generalizability and translation of preclinical trials by embracing both the mean and variability in outcomes.

## Conclusions

We have demonstrated how researchers can quantitatively embrace heterogeneity in phenotypic outcomes with the aim of improving both the replicability and generalizability of animal models. Prior to experimentation, researchers may design their experiments by deliberately selecting methodologies that generate variability in disease state, creating a heterogenous, but broadly representative backdrop of disease states against which treatment efficacy can be assessed [10–12]. Since the magnitude and direction of phenotypic expression and outcomes are determined by the interaction of genetic and environmental contexts within studies [14–16], both of these methodological factors require heterogenization in order to avoid context-specific and non-replicable outcomes across studies [16]. Post-experimentation, studies may further incorporate analyses that estimate the magnitude and direction of variability generated by treatments to identify potentially generalizable versus non-generalizable approaches. Recent meta-analyses of variability in phenotypic outcomes of animal models are beginning to illuminate the potential use of embracing different types of heterogeneities for improving replicability, generalizability, and translation [60–62]. We offer that comparative analyses of variability in both control and treatment groups has the potential to inform experimental design and lead to changes in both the approach and direction of follow-up studies, ultimately leading to a more successful program of replicability, drug discovery, and translation.

## Materials and methods

### Data collection and imputation

We identified studies of rat animal models for stroke from the CAMARADES electronic database (see S3 Fig for database query and selection). For our analysis, we only included experimental studies that reported mean infarct volume (and their associated sample size and standard deviation or standard error) in both control and experimental groups. Where necessary, we calculated the standard deviation from the standard error multiplied by the square root of ($n$– 1), where $n$ is the sample size of the control or experimental group. Furthermore, when a study used multiple treatment groups for a control group (28% of identified studies), we divided the sample size of the control group equally among the treatment groups, which dealt with correlated errors and prevented sampling (error) variances being overly small [64]. Before calculating the effect sizes, we excluded data where (i) the standard error was reported as 0; or (ii) the sample size of the control group when divided was equal to or less than 1. We also excluded categorical predictors that were represented by fewer than 5 data points. Overall, 2.9% and 1.2% of all identified studies were excluded for methodological and drug treatment analyses, respectively (S3 Fig).

For meta-analysis of variance across methodological parameters, we focused on control groups with sufficient group-level information on the methodology of the experiment. Specifically, we collected and coded methodological predictors as closely as possible to the predictors used by Ström and colleagues [37] to produce a comparable meta-analysis (see full model parameters in S1 Table). For meta-analysis of variance across drug treatment, we included data from studies with sufficient group-level information on the drug group, rat strain, and sex of experimental/control groups (see full model parameters in S3 Table). For all analyses, we dealt with missing data via multiple imputation [65,66] using the package *mice* [67] as follows: We first generated multiple, simulated datasets ($m = 20$) by replacing missing values with possible values under the assumption that data are missing at random (MAR) [68,69]. After imputation, meta-analyses were performed on each imputed dataset (as described in the Statistical analysis section), and model estimates were then pooled across analyses into a single set of estimates and errors.

### Calculating effect sizes

For meta-analyzing variance across methodological predictors, we calculated the lnCV and its associated sampling variance ($s^2_{lnCV}$) for each control group. Since many biological systems appear to exhibit a relationship between the mean and the variance on the natural scale (i.e., Taylor's law; [42,70]), an increase in the mean may correspond to an increase in variance. Our data indeed appears to exhibit a positive and linear relationship between log standard variation (lnSD) and log mean infarct volume (S1 Fig). When such a relationship holds in data, it may be most preferable to use an effect size such as lnCV, which estimates variance accounting for the mean, and this is the approach we have taken.

For meta-analyzing variance across drug treatments, we calculated the log coefficient of variance (lnCVR) and its associated sampling variance ($s^2_{lnCVR}$) as given in equations (11) and (12) in Nakagawa and colleagues [69] (S8 Table). When meta-analyzing variance in the presence of Taylor's law as it appears in our dataset, it may be most preferable to use lnCVR (over the log variance ratio, lnVR), which gives the variance of a contrast group accounting for differences in the mean. We therefore report all results using lnCVR in the manuscript. We note that both lnCV and lnCVR assume a linear relationship between log mean and log variance with the slope coefficient of 1 on the log scale. When slope estimates are closer to 0 or

nonlinearities are present in the mean–variance relationship, other metrics of variability such as log variability ratio (lnVR) or an approach that directly estimates the strength of association between log mean and log variance (i.e., an arm-based meta-analysis [69]) based on log SD may be more appropriate (for an example of an arm-based approach, see S9 Table and S4 Fig for galaxy plot of lnRR on lnSD). We advise that future analyses of heterogeneity pick the most appropriate statistic and model of variability based on the mean–variance relationships present in their dataset [71]. In addition to assessing the effects of treatments on variance, we further quantified differences in mean infarct volume by calculating the lnRR of the mean for each control/experimental group within a study (lnRR) and its associated sampling variance ($s^2_{lnRR}$). For both lnRR and lnCVR, we calculated effect sizes so that positive values corresponded to a larger mean or variance in the experimental group.

## Statistical analysis

We implemented multilevel meta-analytic models in a likelihood-based package using the function "rma.mv" in the *metafor* package [72] as described in Eq 1:

$$y_{ij} = \mu + \beta x_{ij} + s_j + t_j + e_{ij} + m_{ij}, \tag{1}$$

where $y_{ij}$ (the $i$th effect size of variability or mean infarct volume from a set of $n$ effect sizes ($i = 1,2,...,n$) in the $j$th study from a set of $k$ studies $j = 1,2,...,k$) is given by the grand mean ($\mu$), the effects of fixed predictors ($\beta x_{ij}$), and random effects due to study ($s_j$), strain ($t_j$), residual ($e_{ij}$), and measurement error ($m_{ij}$) for the $i$th effect size in the $j$th study. Since variability in observed effects may be explained by measurement error ($m_{ij}$ in Eq 1), we present total $I^2$ (the percentage of variance that cannot be explained by measurement error) and study $I^2$ (the percentage of variance explained by study-effects) to estimate the true variance in observed effects (i.e., meta-analytic heterogeneity) [60]. We interpreted $I^2$ of 25%, 50%, and 75% as small, medium, and large variance, respectively [60].

To estimate variance (lnCV) in outcome as a function of methodology in control groups, we constructed 2 meta-analytic models. First, we fitted a multilevel meta-analysis (MLMA) with the objective of estimating the overall average variability in infarct volume across studies. MLMA included a fixed intercept and random effects described in Eq 1. Second, we fitted a MLMR with the objective of estimating effects of methodological predictors on variability in infarct volume, by fitting the following fixed predictors: (i) method of occlusion; (ii) sex of animal cohort; (iii) type of ischemic model; (iv) type of anesthetic; (v) whether experiments were temperature controlled; (vi) whether rats were physiologically monitored; (vii) mean cohort weight; and (viii) time for evaluation of damage after focal ischemia (S1 Table). Mean cohort weight and time for evaluation were z-transformed prior to model fitting. We similarly constructed MLMA and MLMR models for lnRR and lnCVR (fitting each effect size as the response in separate models) to estimate the mean and variance in outcome as a function of drug treatment in our control/experimental groups, respectively. For these MLMR models, we included (i) drug treatment group, and (ii) sex of animal cohort as fixed predictors (S3 Table). Fixed effects were deemed statistically significant where their 95% credible intervals (CIs) did not span zero. For interpretation of results, we back-transformed model estimates from the log to the natural scale.

Since reported outcomes may be prone to within-study biases particularly with regard to mean estimates, we conducted a sensitivity analysis including publication quality as a random effect in our MLMR model. Publication quality was determined according to guidelines set out by the Stroke Academic Industry Roundtable (STAIR), which scored studies based on whether they implemented strategies to mitigate against both selection and detection bias [73].

Our sensitivity analysis did not lead to any qualitative changes in our main reported outcomes, and publication quality accounted for little in terms of differences in mean infarct volume ($I^2_{lnRR}$ = 0.7%; for full sensitivity model estimates, see S5 and S6 Figs and S10–S12 Tables). Finally, we tested for signs of publication bias (systematic bias in the published data due to the preferential publication of more significant results) in our data by visual inspection of funnel plots (S7 Fig) and conducting a type of Egger regression (precision-effect test and precision-effect estimate with standard errors, PET-PEESE) on lnRR [74]. Egger regression on lnRR suggested a small effect of publication bias in our mean estimate (1.6% difference between bias-corrected and uncorrected estimates of our meta-analytic mean; see S13 Table for publication bias test results). Egger regression cannot be used for lnCVR, and further, it is unlikely that publication bias occurs for lnCVR because such biases are not driven by the difference in standard deviations between the experimental and control groups [75]. All meta-analyses were conducted on the statistical programming environment R (v 3.2.2 [76]).

## Supporting information

**S1 Fig. Scatter plot of log mean–variance (log SD) relationship in rat animal data.** Point estimates for control (blue) and treatment (yellow) groups are provided. Slopes and 95% CIs from linear regressions for control (0.822, 0.791 to 0.854) and treatment (0.758, 0.728 to 0.788) rat groups, respectively, are shown. The data underlying this figure can be found at https://doi.org/10.6084/m9.figshare.14527317.v4. CI, credible interval.
(TIF)

**S2 Fig. Relationship between variability (lnCV) induced by occlusion methodologies and consistency (ln$H$) in drug treatment outcomes.** The mean slope of the relationship (slope = −0.876, −2.047 to 0.295) from the MLMR model is shown by the gray line. Circles represent estimates for each occlusion method and solid lines their 95% CIs obtained from multilevel regression (MLMR) models. Each color represents a different occlusion method, and circle sizes represent the number of effect sizes available to estimate ln$H$ for each occlusion method. From the highest to lowest ln$H$ estimates: orange = Filament [$N$ = 973]; yellow = Mechanical/direct [$N$ = 438]; blue = Endothelin injection [$N$ = 76]; turquoise = Emboli/clot [$N$ = 201]; green = Photothrombotic [$N$ = 64]; purple = Collagenase injection [$N$ = 8]; pink = spontaneous [$N$ = 4]. The data underlying this figure can be found at https://doi.org/10.6084/m9.figshare.14527317.v4. CI, credible interval; lnCV, log coefficient of variation; MLMR, multilevel meta-regression.
(TIF)

**S3 Fig. PRISMA flowchart of database query and study selection process.** PRISMA, Preferred Reporting Items for Systematic Reviews and Meta-Analyses.
(TIF)

**S4 Fig. Galaxy plot of treatment effects based on mean efficacy (lnRR) and interindividual variability in efficacy as obtained from an arm-based meta-analysis of lnSD.** Estimates (circles) represent unconditional (marginalized), treatment-specific means (lnRR), variability (lnSD), and their 95% CIs (solid lines). We reveal differences in variability across treatment groups, with treatments that significantly reduce infarct volume and increase interindividual variability (positive lnSD) highlighted blue. The effects of hypothermia and thrombolytics (the latter of which include the only regulatory-approved treatment) are highlighted in pink. The data underlying this figure can be found at https://doi.org/10.6084/m9.figshare.14527317.v4. CI, credible interval; lnRR, log response ratio; lnSD, log standard variation.
(TIF)

**S5 Fig. Sensitivity model CV estimates from multilevel regression (MLMR) of infarct volume in control groups.** Mean estimates of unconditional (marginalized), group-specific coefficients of variation (%) are indicated as gray circles. Moreover, 95% CIs are shown as gray lines and are asymmetric due to back-transformation of log coefficient of variation (lnCV) to the natural scale. The data underlying this figure can be found at https://doi.org/10.6084/m9.figshare.14527317.v4. CI, credible interval; lnCV, log coefficient of variation; MLMR, multilevel meta-regression.
(TIF)

**S6 Fig.** Sensitivity model (a) lnRR and (b) lnCVR estimates from multilevel regression (MLMR) of infarct volume in treatment/control groups. Mean estimates of unconditional (marginalized), group-specific effects are shown as gray circles, and 95% CIs are shown as gray lines. Negative lnRR estimates indicate that mean infarct volume is smaller in experimental versus control rats. Negative lnCVR estimates show that interindividual variability in infarct volume is smaller in experimental versus control rats. The data underlying this figure can be found at https://doi.org/10.6084/m9.figshare.14527317.v4. CI, credible interval; lnCVR, log coefficient of variation ratio; lnRR, log response ratio; MLMR, multilevel meta-regression.
(TIF)

**S7 Fig. Funnel plot for lnRR characterizing differences in mean infarct volume for control/treatment groups.** Raw effect sizes are plotted against their precision (inverse of the square root of standard error). MLMA model predicted mean effect size (solid vertical line), and its 95% CI (dashed lines) are shown. The data underlying this figure can be found at https://doi.org/10.6084/m9.figshare.14527317.v4. CI, credible interval; lnRR, log response ratio; MLMA, multilevel meta-analysis.
(TIF)

**S1 Table. Unconditional (marginalized) estimates and 95% credible intervals for lnCV, obtained from multilevel regression (MLMR) models of control group infarct volume.** Continuous predictors were Z-transformed prior to model fitting. lnCV, log coefficient of variation; MLMR, multilevel meta-regression.
(DOCX)

**S2 Table. Conditional estimates and 95% credible intervals for lnCV, obtained from multilevel regression (MLMR) models of control group infarct volume.** Continuous predictors were Z-transformed prior to model fitting. Bold italicized estimates indicate that the 95% credible intervals do not span zero. lnCV, log coefficient of variation; MLMR, multilevel meta-regression.
(DOCX)

**S3 Table. Unconditional (marginalized) estimates and 95% credible intervals for lnRR and lnCVR, obtained from multilevel regression (MLMR) models of infarct volume in treatment/control groups.** Treatment effects (DrugGroup) are ordered from groups that produce, on average, the greatest reduction in infarct volume (i.e., the most effective, as indicated by most negative estimates of lnRR) to groups that are, on average, the least effective. lnCVR, log coefficient of variation ratio; lnRR, log response ratio; MLMR, multilevel meta-regression.
(DOCX)

**S4 Table. Conditional estimates and 95% credible intervals for lnRR and lnCVR, obtained from multilevel regression (MLMR) models of infarct volume in treatment/control groups.** Bold italicized estimates indicate that the 95% credible intervals do not span zero. lnCVR, log

coefficient of variation ratio; lnRR, log response ratio; MLMR, multilevel meta-regression.
(DOCX)

**S5 Table. Conditional estimates and 95% credible intervals for lnRR and lnCVR, obtained from contrast multilevel regression (MLMR) models to assess the effect of sex on infarct volume.** The intercept here represents studies in which "Both" sexes were used. Bold italicized estimates indicate that the 95% credible intervals do not span zero. lnCVR, log coefficient of variation ratio; lnRR, log response ratio; MLMR, multilevel meta-regression.
(DOCX)

**S6 Table. Conditional estimates and 95% credible intervals for lnRR and lnCVR, obtained from contrast multilevel regression (MLMR) models to assess the effect of sex on infarct volume.** The intercept here represents studies in which only "Female" sex was used. Bold italicized estimates indicate that the 95% credible intervals do not span zero. lnCVR, log coefficient of variation ratio; lnRR, log response ratio; MLMR, multilevel meta-regression.
(DOCX)

**S7 Table. Consistency in drug treatment outcomes across variability induced by occlusion methodologies.** For our second-order meta-regression, we first separated our rat infarct volume data by occlusion methods. For each occlusion method data, we conducted a MLMR to estimate heterogeneity ($I^2$) in lnRR including our original random (study ID, effect size ID, and strain) and fixed effects (sex + drug treatment group). From our MLMR models, we extracted total $I^2$ of lnRR and from this calculated the heterogeneity statistic ln$H$. ln$H$ is a preferable effect size for downstream analyses as it is unbounded and has a relatively well-defined standard error to act as a measure of its precision [61 in main text]. Using the square of the standard error of ln$H$ as the sampling variance and ln$H$ as our response variable, we then fit a second-order meta-regression using the lnCV estimates of each occlusion method as a fixed predictor and effect size ID as a random effect ($\sigma^2_{Residual}$ = 0.200). Unconditional estimates of lnCV were obtained from our MLMR models of methodological variability (S1 Table) described in our main text. Estimates and 95% credible intervals from this second-order MLMR model is reported below. Estimates with credible intervals that do not span zero are considered statistically significant. See S3 Fig for a line plot depicting the relationship between ln$H$ and lnCV with the model fitted line. lnCV, log coefficient of variation; lnRR, log response ratio; MLMR, multilevel meta-regression.
(DOCX)

**S8 Table.** Effect sizes and sampling variances used in meta-analysis of variance (a) across methodological predictors and (b) across drug treatment groups. Equations and the model type in which the effect size was used are also given. $\underline{x}$ and $s$ are the mean and SD of the group infarct volume, $n$ is the sample size, CV is the coefficient of variation, and $\rho$ is the correlation between the mean and standard deviation on the log scale ($\rho$ is assumed to be 0*). Subscripts C and E refer to control and treatment groups, respectively.
(DOCX)

**S9 Table. Model estimates (unconditional) and 95% credible intervals for lnRR and lnSD.** Estimates of lnSD were obtained from an arm-based, multilevel regression model (MLMR) of lnSD in infarct volume for both treatment and control groups. Original fixed (drug treatment group and sex) and random effects (study ID, effect size ID, and strain) were fit, in addition to a nested random effect of "Drug treatment group | pairwise ID." Estimates of lnRR are from the main analysis of mean drug treatment effects and are the same as in S3 Table. Treatment effects (DrugGroup) are ordered from groups that produce, on average, the greatest reduction

in infarct volume (i.e., the most effective, as indicated by most negative estimates of lnRR) to groups that are, on average, the least effective. lnRR, log response ratio; lnSD, log standard variation; MLMR, multilevel meta-regression.
(DOCX)

**S10 Table. Sensitivity model estimates (unconditional) and 95% credible intervals for lnCV, obtained from multilevel regression (MLMR) models of control group infarct volume.** Continuous predictors were Z-transformed prior to model fitting. lnCV, log coefficient of variation; MLMR, multilevel meta-regression.
(DOCX)

**S11 Table. Sensitivity model estimates (unconditional) and 95% credible intervals for lnRR and lnCVR, obtained from multilevel regression (MLMR) models of infarct volume in treatment/control groups.** Treatment effects (DrugGroup) are ordered from groups that produce, on average, the greatest reduction in infarct volume (i.e., the most effective, as indicated by most negative estimates of lnRR) to groups that are, on average, the least effective. lnCVR, log coefficient of variation ratio; lnRR, log response ratio; MLMR, multilevel meta-regression.
(DOCX)

**S12 Table. Sensitivity model estimates of heterogeneity ($I^2$) for analyses of methodology on variability (lnCV) and drug treatment on mean (lnRR) and variance (lnCVR) in rat infarct volume.** Estimates (%) are shown for MLMAs and MLMR models. lnCV, log coefficient of variation; lnCVR, log coefficient of variation ratio; lnRR, log response ratio; MLMA, multilevel meta-analysis; MLMR, multilevel meta-regression.
(DOCX)

**S13 Table. Results from Egger regression (PET-PEESE) on lnRR to test for publication bias.** This procedure fits the square root of sampling variance as a moderator (slope estimate and 95% credible intervals shown in the first half of the table). If this estimate is significant, we then fit the sampling variance (second half of the table). The intercept from this latter model indicates a "potentially" bias-corrected, modified meta-analytic mean. In our case, the biased-corrected estimate is a 28.0% decline, compared with the original estimate without correction, which is a 29.6% decline. These values indicate that although this analysis detected a sign of publication bias, the effect of this bias is very small (1.6% difference). Bold italicized estimates indicate that the 95% credible intervals do not span zero. lnRR, log response ratio; PEESE, precision-effect estimate with standard errors; PET, precision-effect test.
(DOCX)

## Acknowledgments

We would like to thank the CAMARADES team for help in data access and extraction and the I-DEEL lab for providing the opportunity for TU to conduct this meta-analysis. We thank Megan Szojka and Mia Waters for providing feedback on an earlier draft.

## Author Contributions

**Conceptualization:** Takuji Usui, Alistair M. Senior, Shinichi Nakagawa.

**Data curation:** Takuji Usui, Malcolm R. Macleod, Sarah K. McCann, Shinichi Nakagawa.

**Formal analysis:** Takuji Usui, Alistair M. Senior, Shinichi Nakagawa.

**Funding acquisition:** Alistair M. Senior, Shinichi Nakagawa.

**Supervision:** Alistair M. Senior, Shinichi Nakagawa.

**Writing – original draft:** Takuji Usui.

**Writing – review & editing:** Takuji Usui, Malcolm R. Macleod, Sarah K. McCann, Alistair M. Senior, Shinichi Nakagawa.

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
