## [Editor Report · Decision Letter 0]

13 Nov 2020

Dear Dr Usui, 

Thank you for submitting your manuscript entitled "Embrace heterogeneity to improve reproducibility: A perspective from meta-analysis of variation in preclinical research" for consideration as a Meta-Research Article by PLOS Biology.

Your manuscript has now been evaluated by the PLOS Biology editorial staff, as well as by an academic editor with relevant expertise, and I am writing to let you know that we would like to send your submission out for external peer review. Please accept my apologies for the extreme delay incurred while we sought external advice.

Please re-submit your manuscript within two working days, i.e. by Nov 17 2020 11:59PM.

Kind regards,

Roli Roberts

Senior Editor

PLOS Biology

---

## [Decision Letter · Decision Letter 1]

18 Dec 2020

Dear Dr Usui,

Thank you very much for submitting your manuscript "Embrace heterogeneity to improve reproducibility: A perspective from meta-analysis of variation in preclinical research" for consideration as a Meta-Research Article at PLOS Biology. Your manuscript has been evaluated by the PLOS Biology editors, an Academic Editor with relevant expertise, and by two independent reviewers.

You’ll see that although there’s some positivity from both reviewers, each of them suggests some additional analyses, and reviewer #2 suggests that without such analyses the central claims are poorly supported (and without those claims this article is of specialist interest only). There are also questions about how the lessons learned would be applied in real life.

In light of the reviews (below), we will not be able to accept the current version of the manuscript, but we would welcome re-submission of a much-revised version that takes into account the reviewers' comments. We cannot make any decision about publication until we have seen the revised manuscript and your response to the reviewers' comments. Your revised manuscript is also likely to be sent for further evaluation by the reviewers.

We expect to receive your revised manuscript within 3 months. 

**IMPORTANT - SUBMITTING YOUR REVISION**

*Re-submission Checklist*

*Published Peer Review*

*PLOS Data Policy*

*Blot and Gel Data Policy*

Sincerely,

Roli Roberts

Senior Editor,

rroberts@plos.org,

PLOS Biology

REVIEWERS' COMMENTS:

Reviewer #1:

This manuscript examines the sources of heterogeneity in rat animal models of stroke using the CAMARADES database. A discussion on how this meta-analytic approach can be used for assessing treatment efficacy and stability as well as considerations for incorporating sources of variability to improve the utility of preclinical research.

Major points to consider:

(1) A major limitation of this study is it depends on reported outcomes, which are prone to biases. The authors tested for signs of publication bias, which indicated there was a very small bias, however there are also within-study biases (e.g., selective reporting, lack of or unproper randomization, lack of blinding, etc). Although it might be hard to detect the impact these have on the estimates provided like publication bias, it is probably worth noting in the main text the limitation and/or the impacts of within-study biases on this analysis, as well as the results that there appears to be a very small publication bias (which is currently only mentioned in the supplement and methods). 

(2) I agree on the logical exploration of whether sex of the groups used in the experiments affected lnRR or lnCVR, however as Figure 1 illustrates the greatest source of variability was with the occlusion methods (spontaneous in particular). Related to this would be knowing the relationship of methodology and drug treatments (e.g., do occlusion methods of higher variability (figure 1) associate with drug groups that also have higher variability (figure 2?).

(3) Lines 272-274: Maybe by looking at how filament methods vs other methods impact efficacy (e.g., point above about fig 1 and 2) might strengthen this. This naturally would be subsetting the data in a manner where broad conclusions cannot be made, but using it as a case study (e.g., looking at how the use of the filamental approach vs 'non'-filamental approaches impacts the effect of drug treatment(s) on the difference in infarct volume) would help connect how variations in methodology lead to variations in drug treatment effects. This can also serve as a good example that might help the reader translate the findings from this study into their specific area of interest (related to point 4 below).

(4) I would recommend the authors expand their discussion/conclusions about how to incorporate heterogeneity into the design of experiments. The authors comment on the challenge in doing so (e.g., it is not ethically or practically possible to include all possible combinations), but leave the reader still having to navigate what the 'best practice' should be (or maybe more accurately how to utilize the information of variability in experimental design choices). That is, since new studies will always start highly constrained (e.g., specific animal model choices, specific methodological choices, etc), but need to incorporate more heterogeneity among the landscape of possibilities what considerations should a researcher consider outside selecting methodologies that generate variability. What would be undesirable is avoidance of high variability experimental designs, despite including more methodological heterogentiy, that are the most likely to be translatable. Related, it would be undesirable to include many experimental design choices early, exhausting resources and presenting ethical dilemmas, when only a few would be beneficial to purse earlier in the research process (and note: this is why sharing all outcomes are vital - not just positive findings). A paper the authors might consider related to this topic (e.g., the interplay between exploring the space of possible experimental combinations and how it impacts the constraints on reproducibility and generalizability): https://journals.plos.org/plosbiology/article/comments?id=10.1371/journal.pbio.3000691

Minor points:

Line 327 - Fig 3 not Fig 4

The authors should include the number/percent of studies excluded (e.g., standard error was 0, small sample size, etc). A flow diagram (e.g., PRISMA) might be useful here.

Reviewer #2:

This is a well written and important paper that tries to make the case that limiting variability, while potentially increasing statistical power for an individual experiment, is detrimental when one considers the replicability of findings. The paper presents a meta-analysis of preclinical stroke studies, and examines how the variability of outcome depends on both the method for stroke induction and the treatment given.

I agree with the authors arguments fully. The argument for the small sample sizes that are the norm in preclinical research is that the greatly reduced inter-individual variability drastically increases the power to detect experimental effects. This is true and may in some circumstance be justified (e.g. when one is trying to identify specific functions of individual circuits), however as soon as one begins to undertake research with clinical translatability there is a need for findings that generalise beyond tightly constrained boundaries. 

While I agree with this point of view, the main issue I have is that I don't think the paper's analyses fully support these arguments. For example, the authors show that various methods of stroke induction differ in the variability of lesion volume induced but they do not provide any analyses supporting their argument that greater variability is of benefit. This is highlighted when in the discussion (lines 270-274) the authors highlight similar research that drew the opposite conclusion from the same data. 

I would hope that given the large number of studies available to the authors it might be possible to build on the current analyses to give some quantitative weight to the main arguments of the paper. For example, might it be possible to examine if stroke induction methods with a greater CV (e.g. embolic) show greater consistency in terms of treatment effects than methods with low CV (e.g. filamental). Could one perform a meta- regression: (mean CV stroke induction method CV) ~ I2 (for lnRR or a SMD), i.e. each data point would be a different stroke induction method (ideally a separate meta regression for each treatment type) ? If this isn't possible please do explain why and also discuss how one might explicitly test this. Without this kind of analysis the paper's title does not really represent the content and the paper would be better titled and presented as something of interest to preclinical stroke researchers rather than something of potentially broad relevance.

A few other points:

1. Particularly with clinically relevant behavioural outcomes many outcome measures are not ratio scaled. This can be the case where the outcome measure does not have a zero limit or even with data that does have a zero limit, the mean-sd relationship often become nonlinear toward zero. This has led to erroneous conclusions when using the CVR (see the recent retraction of the Maslej et al 2020 in JAMA Psych). The current data looks like the assumption may be met but checking to see that the intercept is not significantly different from 0 for the SD~mean regression would be a good double check)

2. Related to the above point, the increase in CV for most of the treatments is I expect driven by reduction in mean volume for these treatments, as implied by figure 4 in which there it is clear that studies with the more effective treatment show the greater lnCVR. If there is not an almost perfectly linear relationship between SD and mean, then I worry that the same issue that affected the recently retracted Maslej et al paper may be in play here and wonder if the hierarchical model with ln SD as the response might be preferable (the initial analysis using endpoint treatment scores suggested greater CV with treatment due to the fact that treated groups have lower endpoint scores, a reanalysis of this data using the hierarchical model with ln SD as the response subsequently showed that variability did not differ between groups). 

3. Is the lower CV with male and females combined just due to larger samples (if we have two identical samples, the CV of the combined sample will be smaller than the CV of the individual samples)?

4. Minor point: 'Reproducibility' has been defined as "re-performing the same analysis with the same code using a different analyst", while the current study is generally discussing what is often termed 'replicability': "re-performing the experiment and collecting new data". Patil, P., Peng, R. D., and Leek, J. (2016). A statistical definition for reproducibility and replicability. bioRxiv. doi: 10.1101/066803

---

## [Decision Letter · Decision Letter 2]

28 Apr 2021

Dear Dr Usui,

Thank you for submitting your revised Meta-Research Article entitled "Embrace heterogeneity to improve reproducibility: A perspective from meta-analysis of variation in preclinical research" for publication in PLOS Biology. I've now obtained advice from the original reviewers and have discussed their comments with the Academic Editor. 

Based on the reviews, we will probably accept this manuscript for publication, provided you satisfactorily address the remaining points raised by the reviewers. Please also make sure to address the following data and other policy-related requests.

IMPORTANT: Please attend to the following;

a) Please could you make your title more declarative and informative? We suggest "Meta-analysis of variation in preclinical research suggests that embracing variability would improve both reproducibility and generalizability."

b) Please attend to the remaining requests from reviewer #2. If you agree with her/his comment about reproducibility vs replicability, please also propagate this change to the title.

c) Please address my Data Policy requests further down. Specifically, please supply numerical values underlying Figs Figs 1, 2AB, 3, S1, S2, S4, S5, S6AB, S7, and cite the location of the data clearly in each relevant Fig legend.

We expect to receive your revised manuscript within two weeks. 

*Published Peer Review History*

*Early Version*

Sincerely,

Roli Roberts

Senior Editor,

rroberts@plos.org,

PLOS Biology

DATA POLICY:

Regardless of the method selected, please ensure that you provide the individual numerical values that underlie the summary data displayed in the following figure panels as they are essential for readers to assess your analysis and to reproduce it: Figs 1, 2AB, 3, S1, S2, S4, S5, S6AB, S7. NOTE: the numerical data provided should include all replicates AND the way in which the plotted mean and errors were derived (it should not present only the mean/average values).

DATA NOT SHOWN?

REVIEWERS' COMMENTS:

Reviewer #1:

I reviewed the revised manuscript and the responses from the authors. They addressed all of my concerns and I think the manuscript is much improved. I think the paper is now acceptable for publication.

Reviewer #2:

Please add the p value for the slope of the lnCV-lnH meta regression to the main text

Also unless it the meanings are different in this field shouldn't all instances of 'reproducibility' be changed to 'replicability' - they can be taken to mean quite different things

---

## [Editor Report · Decision Letter 3]

4 May 2021

Dear Dr Usui,

On behalf of my colleagues and the Academic Editor, Isabelle Boutron, I'm pleased to say that we can in principle offer to publish your Meta-Research Article "Meta-analysis of variation suggests that embracing variability improves both replicability and generalizability in preclinical research" in PLOS Biology, provided you address any remaining formatting and reporting issues. These will be detailed in an email that will follow this letter and that you will usually receive within 2-3 business days, during which time no action is required from you. Please note that we will not be able to formally accept your manuscript and schedule it for publication until you have made the required changes.

PRESS: We frequently collaborate with press offices. If your institution or institutions have a press office, please notify them about your upcoming paper at this point, to enable them to help maximise its impact. If the press office is planning to promote your findings, we would be grateful if they could coordinate with biologypress@plos.org. If you have not yet opted out of the early version process, we ask that you notify us immediately of any press plans so that we may do so on your behalf.

Thank you again for supporting Open Access publishing. We look forward to publishing your paper in PLOS Biology. 

Sincerely,

Roli Roberts 

Roland G Roberts, PhD 

Senior Editor 

PLOS Biology